# Mapping Construction Grade Sand: Stepping Stones Towards Sustainable Development

Ando Shah*
Suraj R. Nair*
ando@berkeley.edu
suraj.nair@berkeley.edu
University of California, Berkeley
Berkeley, California, USA

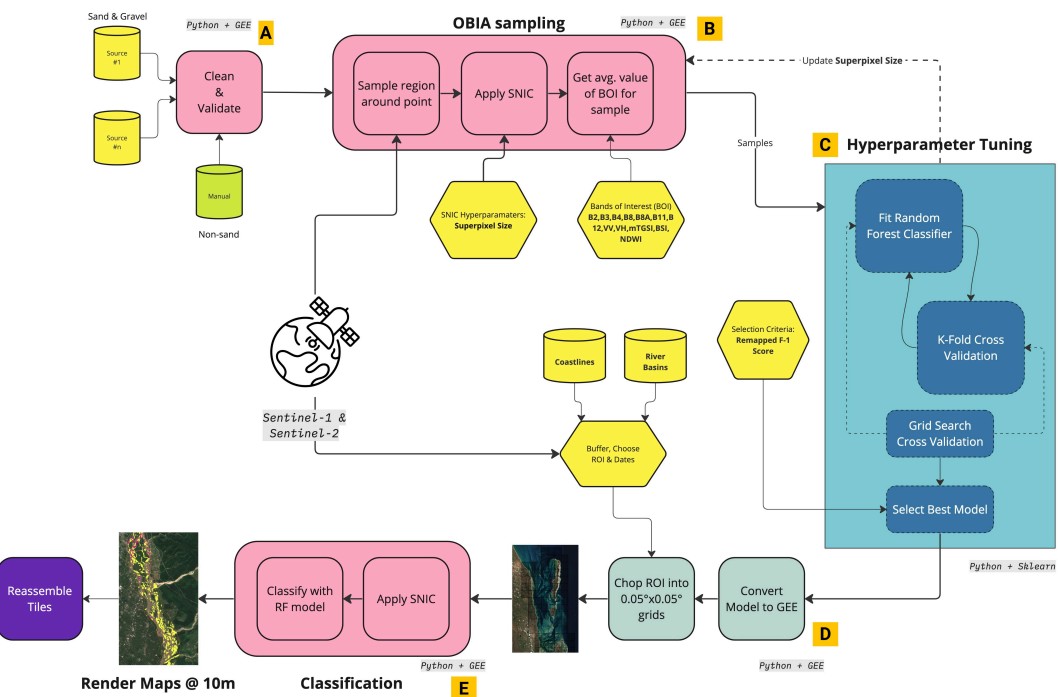

**Figure 1: Model framework for global sand and gravel detection with Google Earth Engine**

## ABSTRACT

Sand and gravel are critical inputs to economic growth as the primary constituents of concrete and asphalt. While demand for these materials has skyrocketed due to large construction and reclamation demands, rates of extraction are unsustainable and result in adverse environmental and socio-economic consequences, especially in the Global South. Excessive sand and gravel mining threatens biodiversity and hydrological functions, heightens the risk of damage to critical infrastructure, and increases vulnerability to extreme climatic events. In this paper, we argue that mapping the world's sand and gravel resources is the first step towards informing effective policy that can ameliorate these harms while achieving sustainable development. We have developed flexible machine learning algorithms which can detect construction-grade sand and gravel resources in river basins and coastlines at global scale with high spatial resolution (10 m). Our approach uses object based image analysis methods fusing freely available Sentinel-1 and Sentinel-2 multispectral satellite datasets. This method achieves an F1 score of 87.5% and accuracy of 88.71% using a random forest classifier trained on a newly aggregated global dataset of in-situ grain size observations. We further validate performance in sections of the River Ganga where a gravel to sand transition is known to occur,

*Both authors contributed equally to this research.

*KDD 2023, Long Beach, California, USA,*

© 2023 Association for Computing Machinery.
ACM ISBN 978-x-xxxx-xxxx-x/YY/MM...$15.00
https://doi.org/10.1145/nnnnnnn.nnnnnnn

and in a section of the River Sone where a number of known sand mining concessions exist. This work lays the foundation to build end-to-end deep learning models that can predict where illegal sand mining occurs.

## CCS CONCEPTS

• **Information systems** → **Geographic information systems**; • **Computing methodologies** → **Object detection**.

## KEYWORDS

machine learning, remote sensing, illegal mining, sand mining, climate change, conservation, infrastructure

**ACM Reference Format:**
Ando Shah and Suraj R. Nair. 2023. Mapping Construction Grade Sand: Stepping Stones Towards Sustainable Development. In *Proceedings of August 6th-10th, 2023 (KDD 2023)*. ACM, New York, NY, USA, 8 pages. https://doi.org/10.1145/nnnnnnn.nnnnnnn

## 1 INTRODUCTION

Sand, gravel, crushed stone and aggregates[1], hereby referred to as sand and gravel resources (SGR), are the most extracted resource on the planet after water, exceeding even biomass and fossil fuels[35]. Sand plays a critical role in delivering ecosystem services, supporting economic development, providing livelihoods, supporting hydrological functions and maintaining biodiversity[35]. Most of the mined material is used for construction (concrete and asphalt are both 80% sand and gravel[2]), land reclamation & flood protection, the need for which is increasing with increasing climate damages associated with higher storm surges, increasingly severe rainfall patterns and sea level rise.

Driven by urbanization and population growth, annual consumption is expected to grow from 24 to 55 gigatons by 2060 [31]. Due to the high costs of sand transport, much of the demand is typically met locally, leading to over-exploitation [31, 35] often in ecologically sensitive areas. The rapid growth in demand is likely to trigger socio-economic conflicts [25]. Excessive sand mining also results in significant environmental impacts, including coastal and river erosion, shrinking deltas, land-use changes, air pollution, salinization of coastal aquifers and groundwater reserves, threats to freshwater and marine fisheries and biodiversity[21, 31].

The primary use of SGR is in the production of concrete, asphalt and glass. SGR is mined by a wide range of actors from large formal companies, to informal artisanal and small-scale miners who often mine in circumstances of poverty [30]. While good governance is integral to the managing these challenges, sand mining is *unregulated* and *under-regulated* in most parts of the world. As a result, sand actors have exploited the absence of regulation and oversight to control markets through coercion and violence [26]. The human costs are faced by miners and local communities who risk drowning, subsidence and landslides, amongst other hazards. Improved monitoring and regulation is thus the need of the hour.

A key impediment to effective regulation, monitoring, and enforcement is the lack of reliable, up-to-date data; as a result, we know very little about where SGR are deposited and extracted. Indeed, one of the primary recommendations[3] of the United Nations Environmental Protection Agency (UNEP) [35] is to produce "comprehensive knowledge on occurrence and distribution, composition, and dynamics, in combination with environmental impact of extraction, is therefore critical for developing long-term strategies for optimised resource use". Mapping the world's sand and gravel resources is thus the first step to inform effective policy that can ameliorate these harms while achieving sustainable development. By mitigating the negative impacts of sand mining through advanced detection techniques, this work can help positively affect the Sustainable Development Goals (SDG) related to environmental protection on land (SDG 15), responsible consumption and production (SDG 12), clean water and sanitation (SDG 6), and sustainable cities and communities (SDG 11).

By bringing together recent advances in object-based machine learning and satellite sensor technology we propose a new method to automatically map the extent of SGR deposits globally, allowing for effective monitoring of these resources. Previous studies have either focused on quantifying percentage of sand in soil worldwide at very coarse resolutions ($\geq 250m$) [10], or quantify the extent of sand in coastal beaches[16] without any notion of the usability of that sand. Our method makes a substantial improvement in this regard; we design a multi-spectral ML model capable of distinguishing between sand, gravel (as defined by the UNEP-GRID recommendations[4]) and all other land use classes from freely available 10m Sentinel imagery[5]. We combine image based object analysis methods with a random forest (RF) classifier trained on a global dataset of sand and gravel grain size, to accurately discern between sand, gravel and other land cover types at 10m resolution. We provide public access to Google Earth Engine (GEE) files that allow the general public to generate predictions for SGR distributions in their regions of interest, without the need for expertise in building and deploying resource-hungry deep learning pipelines.

Finally, we demonstrate the performance of this model for SGR across a large spatial and temporal range of images, with minimal human input. We show that the proposed methods can be used to analyze the impacts of sand mining policies over any given period. The novel contributions of this work include: the development of a multi-spectral machine learning model that utilizes optical-synthetic aperture radar (SAR) fusion, a worldwide dataset of geocoded, dated and verified SGR deposits with known grain sizes, a full processing pipeline for monitoring SGR via spatial and temporal mosaicing, a sensitivity analysis of the most important bands for SGR classification, and case study maps.

## 2 RELATED WORK

The studies that estimate either the grain size directly in fluvial or coastal settings [4–6, 12, 17, 27, 37] typically focus on data collection and modeling of very specific traits of interest such as grain-size

---

[1]Aggregate is a granular material of natural, processed, or recycled origin used essentially for construction purposes[33]

[2]Construction-grade sand must be of a particular grain size (0.075-4.5 mm) and certain coarseness, available only in riverbeds and coastlines, making widely available desert-sand useless for this purpose

[3]Recommendation 6 : Map, monitor and report sand resources

[4]There is a large variance in grain size requirements for the production of concrete and asphalt globally, and the UNEP-GRID have devised a harmonized system with a working definition for both sand and gravel

[5]provided by the European Space Agency

transitions, beach face slope characteristics, etc that make them unsuitable to generalization. Ren et al. [24] employ a combination of remote sensing imagery and locally collected covariates encompassing hydrodynamic and bathymetric attributes of the sand banks to generate predictions. Moreover their high resolution model is sensitive to small spatial changes in those covariates, and to our best knowledge, such data do not exist for most of the world's river basins. Other studies conducted by Marchetti et al. [18] use high resolution drone imagery in subsets of their study regions to generate texture maps for the entire regions of interest. This enables them to correlate remote sensing data to the generated texture and observed grain size labels with machine learning in order to create high resolution grain size maps for the Po river valley in Italy. Chen et al. [2] use convolutional neural networks (CNN) on drones images of gravel beds to predict grain sizes in similar images without a mapping effort. These methods would be difficult or impossible to adapt to the task at hand due to their lack of scalability or dependence on difficult-to-collect local variables. However they provide a strong support for a set of hypotheses that this work builds upon.

An important underpinning is the finding by Pilorget et al. [22] that the single scattering albedo of smaller particles is greater than that of larger ones, which was found in laboratory studies . These studies suggest that in the visible and near-infrared region, even a small variation in the size of a mixture of particles with a given composition and scattering properties tends to control the overall photometric behaviour[18]. Other high-effort, limited-area studies indicate that the percentage of fine sand modulates the reflectance over a large spectra from visible to shortwave infrared wavelengths specifically for sand and gravel deposits[13, 17, 41]. Additionally, van der Wal and Herman [34] find that the thermal infrared (TIR) and C-band backscatter correlate with the median grain size of tidal sediments in the Netherlands. Purinton and Bookhagen [23] find that C-band SAR backscatter provides correlation with sand grain size variability in certain river valleys in Argentina. Given that these studies were intensively conducted in small regions with a focus on local variations, and that their coefficients vary widely, it is not clear if these correlations will hold globally. A contribution of this study is to assess the hypothesis that passive visible, NIR, SWIR, TIR bands and active SAR can be predictors of grain size for sand and gravel resources at a global level. Furthermore, since the SWIR bands are likely to have an impact on class separation, it is also hypothesized that indices such as bare soil index (BSI) that are tuned towards detecting soil characteristics will play an important role in classification task.

The topsoil grainsize index (TGSI) was introduced by Xiao et al. [40] to provide a index that would correlate with the grain size of the topsoil. This method was employed to see if there was a correlation with the grain size of sand, or more correctly, the class predicted. This index is described as:

$$TGSI = \frac{R-B}{R+G+B}$$

where:
$$R = RedBand, G = GreenBand, B = BlueBand$$

**Table 1: Grainsize Database Information**

| Author | Year of study | Geography | Grain Size ? | Geo-locations? | Usage |
|---|---|---|---|---|---|
| Marchetti et. al | 2018 | River Po, Italy | Yes | Yes | Test/train |
| Wilkerson and Parker | 2011 | Global | Yes, only D50 | Sometimes | Test/train |
| Buscombe | 2020 | Coastal USA | Yes | Yes | Test/train |
| Florida FWCC | unknown | Florida, USA | No, only fine, medium, coarse, gravel | Yes | Test/train |
| Knight et. al | 2022 | Oyster Bay, South Africa | Yes | Yes | Qualitative analysis |
| Dingle et. al | 2015-2022 | India, Nepal | Yes | Yes | Test/train |
| Mozambique concessions | 2014-2023 | Mozambique | No | Yes | Qualitative analysis |
| India concessions | 2022 | Multiple states, India | Sometimes | Yes | Qualitative analysis |
| Trampusch | 2005-2018 | Global | Yes, only D50 | Sometimes | Test/train |
| Singh et. al | 2020 | Kashmir, India | Yes, only D50 | Yes | Test/train |
| Quick et. al | 2019 | India, Nepal | Yes | Yes | Test/train |
| Shah and Nair | 2023 | Mozambique | Yes | Yes | Test/train |

## 3 DATA AND METHODS

We assemble and contribute to a high quality geo-located dataset on grain size of sand and gravel. We identify and obtain data from a number of studies where grain size and accurate geo-coordinates were collected, and made publicly available. Examples include Buscombe [1] for coastal measurements in the greater United States, and Dingle et al. [4, 5, 6, 7] for data grain size transitions (GST) across Himalayan rivers[6]. We also include data from Marchetti et al. [18], Wilkerson and Parker [38], Trampush et al. [32] and the Florida Fish and Wildlife Conservation Commission's beach survey database[7] in our analysis.

Although Knight and Abd Elbasit [13] provide more than 300 geocoded grain size data points in Oyster Bay, South Africa, we only use a small subset of this data for validation[8]. Furthermore, many authors do not explicitly provide locations and/or geo-coordinates[14, 17, 37] for riparian sites. Data from Yu et al. [41], Lang et al. [14], Mendes Silveira [20] and the Massachusetts Beach Grain Size and Slope Data[39], Hoque et al. [11] and Ren et al. [24] were not mined at the time of writing and are likely to yield many more important data points. An initial investigation into these sources showed that they only contained data on the sand class, and almost none on gravel, which is the class with the least amount of labels; we therefore concluded our data mining process at this juncture. We

---

[6]While a large literature examines GSTs, most studies lack data grain size or location information[8, 12].

[7]Database of beaches in Florida by the Florida Fish and Wildlife Conservation https://gis.myfwc.com/Data/KMZ_files/

[8]This is due to the fact that this dataset was collected within an area less than one square kilometers on the same day, leading to high probability of spatial autocorrelation between datapoints.

also considered national and sub-national databases on sand mining concessions, such as the states of Uttar Pradesh, Bihar, West Bengal, Madhya Pradesh in India[9], the countries of Mozambique, Tanzania, Ethiopia, Ivory Coast and Uganda[10], for training; since these databases do not contain grain size information, these will only be used for validation and qualitative analysis. Finally, we conducted field studies to gather data on sand grain size and location along three separate stretches of coastline in southern and central Mozambique (where heavy-sand mining sites have been established) and along the Incomati river basin in Mozambique (where a large number of active sand mines are situated).

We summarize the full dataset used in this paper in Table 1. It is important to note that not all locations had exact sampling dates recorded or the sampling dates happened to be prior to 2017 (which is the earliest Sentinel-1 and Sentinel-2 harmonized datasets were available in GEE). For those labels, we used Google Earth Pro to identify a timestamp when those sediment deposits were visible, and not eroded and or submerged.

We have developed a new spectral index, termed the "modified topsoil grainsize index" (mTGSI), which expands on the topsoil grainsize index (TGSI)[40] to include the SWIR and NIR bands. We find that adding these bands to the formula yielded better results for separability of sand v/s other cover types. This index has been termed the 'modified topsoil grainsize index (mTGSI)'. We use this index as an additional feature along with the raw bands. Formally, we define this index as follows:

$$mTGSI = \frac{R-B+SWIR2-NIR}{R+G+B+SWIR2+NIR}$$

where bands correspond to the following Sentinel-2 bands:
$$R = B4, G = B3, B = B2, NIR = B8, SWIR2 = B12$$

Additional bands such as SWIR1 and TIR were considered for this index. The addition of SWIR1 did not produce any qualitative benefits during the label selection stage in GEE and was not pursued. TIR would likley have been a valuable addition, but is not available at 10m resolutions from any freely available sources at the temporal frequencies considered.

Fig 1. describes the model framework and workflow, and the experimental design of this project. The imagery chosen for this research was sourced from the ESA's Sentinel-2 Multi-Spectral Instrument and the Sentinel-1 C-band SAR datasets using GEE Python API to access their Cloud Optimized GeoTIFFs.

For every location in the aggregated database, we validate the data in the satellite imagery manually, to ensure that the location still has SGR deposits at different timestamps. Given the dynamic nature of fluvial and coastal deposits, it is not uncommon for a sand bank/bar/coastline to have drastically altered since in-situ sample collection. For validated samples, an object-based image analysis (OBIA) workflow is applied on the region of interest (ROI) centered on the final position of the label over a region of size 10km x 10km (Fig. 1b). This produces a set of simple non-iterative

clusters (SNIC)[29], and the median of the band values of interest for the cluster/object containing the sample are recorded for that label. A set of non-sand classes were identified and labeled to effectively capture nuances in both riparian and coastal sites. These non-sand classes were arbitrarily selected from across the globe (usually where the authors had personal experience of having visited or lived) from sites that were very clearly representative of the underlying class. These seven classes are: sand, gravel, water, white-water, green vegetation, bare/impervious/developed and cobble. Here cobble refers to the class of sediment that has a larger grain size than gravel (i.e. >75mm median grain size). Early experiments with fewer classes led to significant mis-classifications and motivated the introduction of additional classes such as 'whitewater', which is useful for coastal sites. This proposed classification pipeline classifies the seven identified classes, but collapses them to only three classes of interest, post-classification - 'sand', 'gravel' and 'other'. We also conduct experiments whereby we collapse our data to these three classes prior to training. Doing so allows us to understand if the model is able to better distinguish between classes of interest if all classes are provided, or if the 'other' category is mixed between sub-cover types. Moreoever, it allows us to conduct a more accurate sensitivity analysis of choice of bands on the classification outcome, since we eventually only care about separability between the three classes of interest, and not the separability between all cover types. This knowledge can help downstream projects, such as sand mine detection, to reduce computational load by prioritizing only the high value bands, and is an additional contribution of this work.

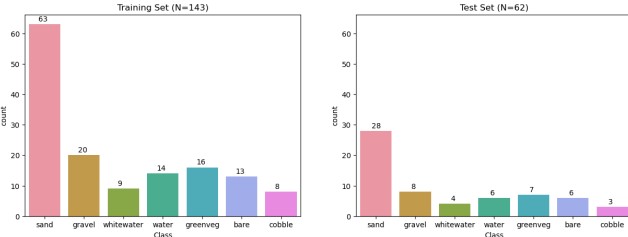

**Figure 2: Train/test split and class distribution**

This problem is framed as one of land-use classification, whereby a RF classifier will try to minimize errors between classified pixels and labels, as shown in Fig 1c. This choice was motivated by prior experience of RFs producing high quality classification for landuse applications. The dataset is split into a training and test test with a 70% (train: N=143): 30%(test: N=63) split, conducted in a stratified manner such that both sets will have an identical distribution of classes as shown in Fig. 2. From the same figure, the imbalanced nature of classes is evident. The number of labels in the non-sand classes in totality are roughly equal to the labels in sand and gravel classes combined; however, the gravel class is highly under-represented which causes issues in classification. This was due to the unavailability of more high quality geo-located data on gravel deposits worldwide and more data collection would needed to ameliorate this issue. Therefore stratification during splitting into train and test buckets, as well as in the cross-validation step

---

[9]There is no central database for the aforementioned national and subnational mining concession reports and they were located by scraping state websites manually. Though most documents contain polygons of concessions, they were not in a machine readable format and considerable effort was and will be expended in digitizing them.
[10]The links to the national databases can be found at https://landadmin.trimble.com/cadastre-portals/; not machine readable.

are crucial in ensuring generalizability to held-out datasets. We use the Sklearn package in Python to fit a RF classifier with a range of hyperparameters, which we then tune using the grid search along with a 5-fold cross validation procedure [11]. A modified F1 score was chosen as the 'refit' function (henceforth referred to as the 'remapped F-1 score'), whereby the outcome classes are first remapped to just 'sand', 'gravel' and 'other' and then the F1 score is calculated as a weighted average for only the sand and gravel classes, thereby selecting the model that maximizes this score. F1 scores are defined as:

$$F1 = \frac{2*Precision*Recall}{Precision+Recall} \text{ where } Precision = \frac{TP}{TP+FP} \text{ and}$$
$$Recall = \frac{TP}{TP+FN}$$

Here TP, FP and FN are the number of true positives, number of false positives and number of false negatives, respectively. We focus on the F1 score in order to achieve a balance between false positives and false negatives. Consider the scenario where optimal policy aims to reduce over-extraction of existing SGRs; here, minimizing false positives becomes important. On the other hand, when policy aims to enable new mining concessions for under-extracted SGRs, minimizing false negatives becomes more important.

During this process, the hyperparameters for RFs as well as for the SNIC algorithm are tuned in this step. The only SNIC hyperparameter used was the size of the superpixel, or the grid size from where the superpixels are grown, as this is known to have an outsized effect on the outcome[29]. The penultimate step (Fig. 1e) involves applying the OBIA workflow again to first generate SNIC clusters and then classify them into one of seven classes. The inference step described above is conducted on particular sections of interest over dates of interest. For example, for river basins in India, a flood extent calculation was conducted using Sentinel-1 data to ascertain when the river has the least water, thereby exposing the most amount of SGR. A median of Sentinel-1 and Sentinel-2 datasets over 5 images is constructed around a date of interest to minimize cloud cover. The classification step is then performed over this composite image and is further limited to region of interest around a river basin. We typically use a buffer of 1 km around the centerline of major rivers.

## 4 RESULTS

The experiments carried out in this study aim to address two key aspects of the research. First, we validate the predictions of the machine learning model, so that it can be used to benchmark more sophisticated techniques in the future. Second, we assess how it could be used to implement policy based on its predictions.

The best performing model achieved an F1 score of **87.5%** (weighted average for sand and gravel classes only) and an overall accuracy of **88.71%**. These correspond to a superpixel size of 10 pixels or 100 meters in real space for the SNIC algorithm. For the best performing model, the confusion matrices for both the original set of land-use classes as well as the final remapped classes are shown in Figs. 3a) and 3b), respectively. The model performance on the global held-out test set shows a relatively satisfactory level of performance for the

[11]The Sklearn RF hyperparameters tuned were: 'bootstrap'(True/False), 'max_depth' (50-250), 'n_estimators' (1000-3000), 'min_samples_leaf' (1,2,4), 'min_samples_split' (2,3,4,5)

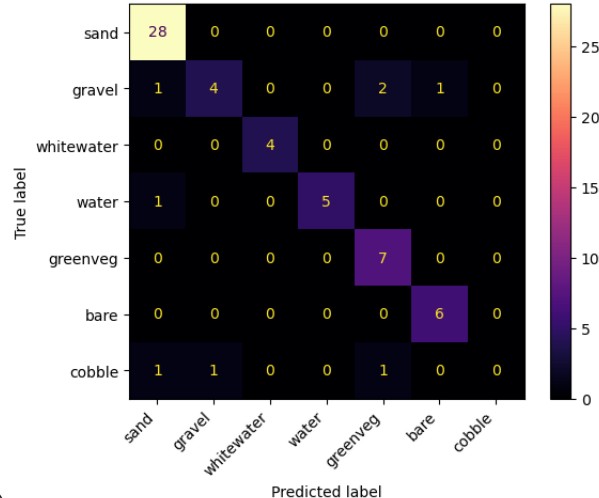

(a)

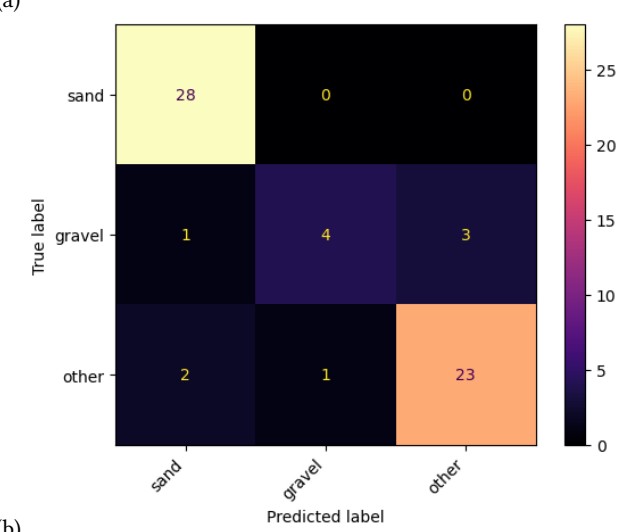

(b)

**Figure 3: Confusion matrices evaluated on the held-out test set for the best performing random forest classifier for: (a) Original 7-class classification (b) Remapped 3-class classification**

sand and gravel classes, which are very much in line with existing studies that use RFs for land-use classification[15, 29].

We also conducted experiments where the classes were first collapsed to the three of interest, i.e 'sand', 'gravel' and 'other' by mapping all non-sand classes to 'other', and subsequently trained an RF model using the same procedure. This model did not outperform the 7-class model, but a feature importance analysis was conducted on the held-out test set using this model as show in Fig. 5.

### 4.1 Grainsize Transition Analysis

A large body of work on fluvial systems highlights the sharp transition from gravel to sand within a short distance[5, 8, 36]. As sediment is carried downstream by rivers, particles get finer in the absence of material input; once the median grain size reduces to ∼10

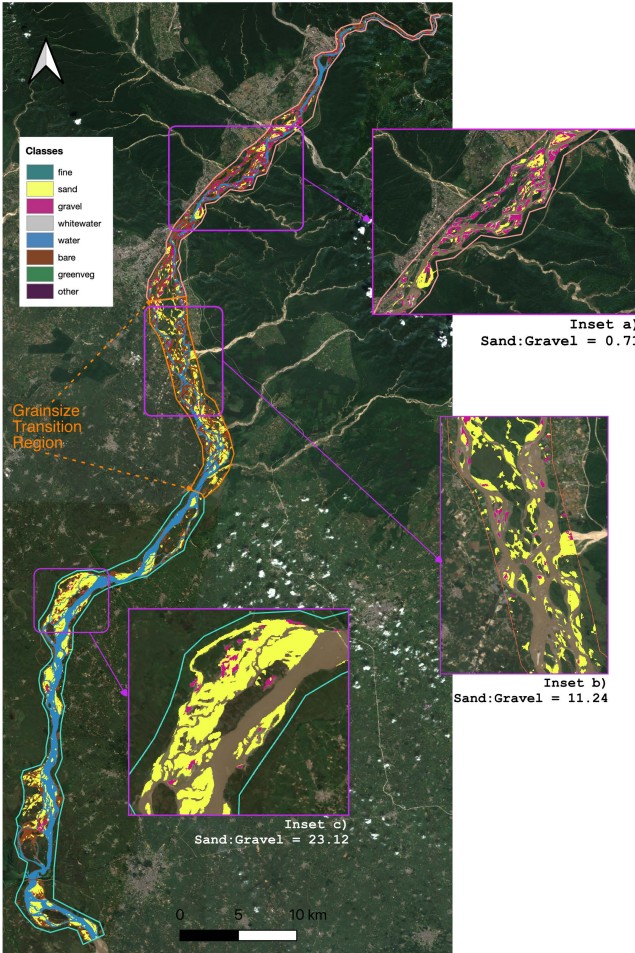

**Figure 4: Section of the River Ganga where a sharp grain size transition (GST) is known to happen. Inset a): upstream of the GST, where the gravel dominates; Inset b): within the GST where there is a sharp increase in the sand to gravel ratio; Inset c): downstream of the GST, where sand to gravel ratio increases substantially.**

mm, there is an abrupt transition to sand. This transition is termed the gravel-sand transition (GST) and can occur over distances as little as a few hundred meters[5, 36]. Here one such river system has been analyzed as shown in Fig. 4, the river Ganga in northern India, as it comes down from the Himalayas and experiences the GST. The river has been described in three sections: a) upstream of the GST; b) within the GST and c) downstream of the GST. These sections were inferred from the data provided in Dingle et al. [6]. The number of pixels representing sand and gravel are calculated for each section using the best model. The sand:gravel ratios in each section are 0.71, 11.24 and 23.12 for the upper, GST and lower sections respectively. From these ratios we see that the sand:gravel ratio increases suddenly in the GST and continues to be dominated by sand deposits downstream of the GST, consistent with Dingle et. al's observations. Unfortunately, no known baselines exist for these

ratios and further in situ validation would be needed. In the absence of which, this localized analysis provides us with a secondary level of confidence in the predictions of the model.

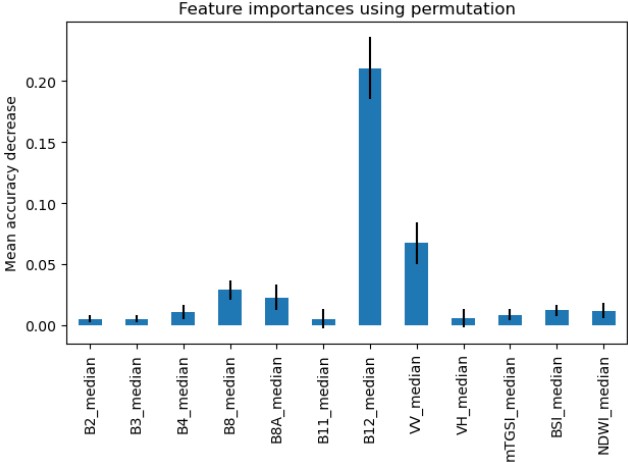

**Figure 5: Feature importance plot for the multispectral RF model**

## 4.2 Active Sand Mining Site Analysis

Figure 6 is a map of sand mining concessions in the state of Bihar in northern India over a small section of the river Sone, and the predicted sand and gravel deposits in that region. The amount of sand deposited is compared in the pre-monsoon and post-monsoon seasons, with the hypothesis that the monsoon will provide replenishment. During the pre-monsoon season it was found that 90.5% of concessions in this area had SGR deposits, whereas during the post-monsoon season this number increased to 100%. This is indicative of a positive replenishment rate responsible for buffering the stock of SGR by depositing sand during the monsoon season.

## 5 DISCUSSION

Despite the promise of using this method to detect SGR at scale, it suffers from numerous challenges. First, the overall accuracy and F1 scores are currently hindered by the confusion between spectrally similar classes. On close analysis of the 7-class confusion matrix shown in Fig 3 a), we see a fair bit of confusion between the green vegetation and gravel classes. This is pointing to a dearth of samples representing the gravel class in the training set (N=20), which we have identified as a challenge. A small amount of confusion also occurs between the cobble and sand/gravel classes, which is perhaps to be expected in terms of class delineation errors and global nature of the dataset, especially given the low number of samples from the cobble class in the training set (N=8). However, these confusions are the main contributors to the reduction of the F1 score, as seen in the 3-class confusion matrix as shown in Fig. 3 b). These can be ameliorated by adding more labels to the dataset, an ongoing effort. We also see that the performance for the sand class is flawless, which was likely boosted by the number of high quality sand samples in the training dataset (N=63).

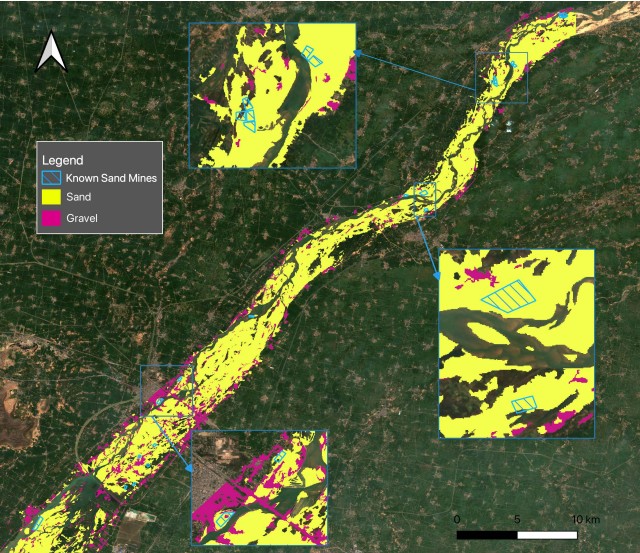

**Figure 6: Known sand mining concession (N=23) and predicted sand and gravel cover types along the Sone river in Bihar, India in the pre-monsoon month of April 2022. Insets show details around selected sand mining concessions.**

Secondly, the highly dynamic nature of SGR deposits, especially for riverbanks is a major challenge for local validation. For example, a validation strategy could be to use the pixels within known sand mining concession as positive labels. However, the inter-annual and intra-annual changes in the hydrological system can wreak havoc with this strategy; pixels within concessions are often inundated with water, or the deposits have changed or migrated since the concessions were allocated.

To test the efficacy of using a multspectral classification strategy, we also trained a model with only the RGB bands. The best performing model with RGB bands only achieves an F1 score and accuracy of 60.61% and 61.22%, respectively; substantially lower than the multispectral model, highlighting the value of multispectral data for this use-case.

We conducted a sensitivity analysis to understand which bands contribute most towards the classification outcome. To do this accurately, we first collapsed the training dataset into 3 classes of interest, where the 'other' class was a remap of all non-sand classes. Subsequently a RF model was training in the same manner as before, and the best performing model was selected (F1 Score for sand & gravel only: 82.07%, overall accuracy: 77.78%). Then we conducted a feature importance analysis based on feature permutation, which does not have a bias towards high-cardinality features (which our band information is likely to have), and computed on the held-out test set. From the feature importance plot of the multispectral model shown in Fig. 5, we can see that the features that contribute most to the result are SWIR2, VV, NIR, Red Edge 4, Red, BSI & mTGSI bands, in order of decreasing contribution. The combination of the multispectral information, both optical and SAR, along with the spectral indices led to higher relative performance, and therefore the use of this information is key to the detection of SGR, and maybe useful for future work.

## 6 FUTURE WORK AND CONCLUSION

Anticipated future work involves incorporating a few different directions. The first is to be able to detect illegal sand mines by learning their spatial and spectral signature from known active mines. Deep learning models that solve semantic segmentation tasks such as convolutional neural networks or vision transformers have proven to be adept at detecting artisanal scale mining operations in satellite imagery[9, 19, 28], and we believe that they will be quite effective at detecting sand mines, given accurate labels. However, we strategically chose to use to not use deep learning methodologies to solve this problem, so that the outcomes could be deployed by anyone with knowledge of the freely-available GEE, simply by using the visualization file that we have provided, as opposed to the rigors and resource requirements of building and deploying deep learning pipelines.

Since our described method produces very lightweight models, their outcomes could be applied as a mask over regions of interest, and known sand mines could be used as positive labels to tune such models. This approach reduces both the labeling burden (knowing where sand exists decreases the search space) and compute demands during inference (by reducing the search space for possible mine locations). This work could also serve as a means to identify at-risk SGR deposits that are either over-extracted, at risk of extraction due to proximity to urban centers, or vulnerable due to proximity to protected areas, and serve as a means to protect them. Therefore this work lays the groundwork for the detection of illegal, unregulated or destructive sand mining at scale. Finally, we show that the use of multispectral information from the fusion of S-1 and S-2 data, along with certain spectral indices are key to detecting SGR, an aspect that could be leveraged by future studies to develop sparser models.

Furthermore, the overall goals of this project may be well served by applying participatory or community design principles. Danielsen et al. [3] provide some guiding principles for best practices, which could lead to greater awareness of the threats of rampant sand mining, optimal policy outcomes, continuously updated models and enhanced trust and buy-in of stakeholders. We imagine a system where anyone can anonymously report instances of sand mining - illegal, informal or otherwise, along with their lived experience. These community inputs could be used to update training labels and provide additional reports of illegal sand mining. One of the key requirements of such community monitoring projects must be the guarantee of the safety and privacy of participants.

In conclusion, our method produces flexible and lightweight machine learning models that can reasonably detect construction-grade sand and gravel at scale globally, using multispectral, open access remote sensing datasets. It lays the groundwork for adaptive and optimal policymaking that may alleviate the burden imposed by the development demands of the Global South on its environment and society.

## 7 DATA AVAILABILITY

The aggregated dataset and code are freely available from this repository: https://github.com/BerkeleySandProject/py-sand-mapping.

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
