# OpenReview forum: "Mapping Construction Grade Sand: Stepping Stones Towards Sustainable Development"
_KDD.org/2023/Workshop/Fragile_Earth — KDD 2023 Workshop Fragile Earth Submission_

### Official Review · Reviewer_YUGs · 2023-07-12
**The paper is well written with a detailed discussion about the Sand and Gravel mining issue . Presenting applicability of results for Gravel Analysis and Sand and Mining analysis is a plus. Limitations and future work is also sufficiently talked about. There is lack of comparison with other existing methods in the land-use segmentation literature. Overall it sufficiently covers solution to an environmental challenge using machine learning method and hence gets my acceptance.**

**Rating:** 7
**Confidence:** 2

**Review:**

-Summary : Environmental challenge of Sand and Gravel extraction has been explored. Problem is presented as an object classification problem, where construction grade sand and gravel can be detected at high resolution using machine learning. Inference and applicability of results from their Random Forest based method has been provided for Sand Grain transition analysis and Active Sand Mining Analysis to aid policy making around the environmental impact of these issues.

- Strengths:
      - a. Detailed analysis of the Sand and Gravel Mining Issue has been presented
      - b. Applicability of their method is discussed in great detail.
      - c. Limitations and Future work has been provided
-  Weaknesses:
      - a. Performance with other methods isn't provided
      - b. Some details such number of images used for training and choice and justification for use of 10-fold Cross Validation isn't
                  provided
-Questions : What are the performance differences between other ML methods such as SVMs and KNN ? What are challenges in deploying ML based methods to solve challenges of Sand and Gravel mining?
-Limitations : More thorough analysis with other classification methods are needed .

---

### Official Review · Reviewer_Ffus · 2023-07-13
**Review for "Mapping Construction Grade Sand: Stepping Stones Towards Sustainable Development"**

**Rating:** 7
**Confidence:** 5

**Review:**

This paper proposes to detect sand and gravel deposits using Sentinel imagery. The paper is focused on a very specialized area where there is an environmental impact on un-regulated sand/gravel mining. Therefore, from an application standpoint, it is well suited for this workshop. In addition, the proposed approach uses a new spectral index, i.e. modified topsoil grainsize index. This new method considers SWIR/NIR bands to further improve the bare soil and sand detection. That being said, from the domain perspective, this paper is well written and considered. However, my primary concern is the choice of the ML method for the task. The existing computer vision methods for land cover classification are way more advanced than the proposed method in this paper. For example, UNet, DLinkNet, etc. image segmentation methods are well established and widely used for tasks such as road, agricultural field, etc. inferences. Moreover, the effort to use these methods is literally plug and play since they are also developed for remote sensing imagery. That being said, I will be happy to see this paper in the workshop but I would like the authors to have a comparative analysis with the existing methods to see how they perform for the task in hand.

---

### Official Review · Reviewer_UHi6 · 2023-07-16
**Review for "Mapping Construction Grade Sand: Stepping Stones Towards Sustainable Development"**

**Rating:** 7
**Confidence:** 4

**Review:**

Summary:

This paper develops flexible machine learning algorithms which can detect construction-grade sand and gravel resources in river basins and coastlines at global scale with high spatial resolution (10 m). The experiment on a large spatial and temporal range of images shows that the proposed methods can be used to analyze the impacts of sand mining policies over any given period.

Strengths:

- The paper is well-written and the motivation is very attractive.
- The core idea is straightforward to implement and the proposed method achieves promising F1 score and accuracy.

Weaknesses:

- It would be helpful if the authors can add a related work section which can help reviewers and future readers better understand the existing works in detecting construction grade sand and gravel.
- I suggest the authors provide more detailed experimental setup.

---

### Decision · Program_Chairs · 2023-07-19

**Decision:**

Accept (Oral)

**Comment:**

Congratulations!

We are pleased to inform you that your submission: Mapping Construction Grade Sand: Stepping Stones Towards Sustainable Development has been accepted to The KDD 2023 Workshop Fragile Earth: AI for Climate Sustainability - from Wildfire Disaster Management to Public Health and Beyond.

Camera ready deadline is ** July 24 AOE **.  Please log in to OpenReview and prepare your camera-ready version based on the reviews. Formatting rules are the same as for the initial submission and submissions must adhere to KDD 2023 guidelines available at https://authors.acm.org/proceedings/production-information/taps-production-workflow.

Again, congratulations on the acceptance of your paper!  We look forward to seeing you at the workshop on Aug 7, 2023.

The Fragile Earth Workshop Proceeding Chairs